# VIBROACOUSTIC FREQUENCY RESPONSE PREDICTION WITH QUERY-BASED OPERATOR NETWORKS

## ABSTRACT

Understanding vibroacoustic wave propagation in mechanical structures like airplanes, cars and houses is crucial to ensure health and comfort of their users. To analyze such systems, designers and engineers primarily consider the dynamic response in the frequency domain, which is computed through expensive numerical simulations like the finite element method. In contrast, data-driven surrogate models offer the promise of speeding up these simulations, thereby facilitating tasks like design optimization, uncertainty quantification, and design space exploration. We present a structured benchmark for a representative vibroacoustic problem: Predicting the frequency response for vibrating plates with varying forms of beadings. The benchmark features a total of 12,000 plate geometries with an associated numerical solution and introduces evaluation metrics to quantify the prediction quality. To address the frequency response prediction task, we propose a novel frequency query operator model, which is trained to map plate geometries to frequency response functions. By integrating principles from operator learning and implicit models for shape encoding, our approach effectively addresses the prediction of resonance peaks of frequency responses. We evaluate the method on our vibrating-plates benchmark and find that it outperforms DeepONets, Fourier Neural Operators and more traditional neural network architectures.

Code and dataset: https://anonymous.4open.science/r/FRONet-5536

## 1 INTRODUCTION

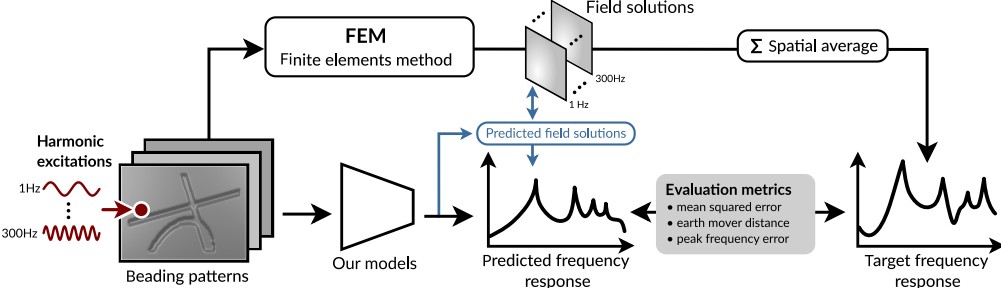

Figure 1: We introduce a dataset of 12,000 samples for predicting frequency responses based on plate geometries. A harmonic excitation with frequency $f_i$ from 1 to 300 Hz is applied to all plates at a fixed location, causing them to vibrate. The velocity field of the vibrating plates (field solution) is obtained through FEM and then spatially averaged, resulting in a frequency response over $f_i$.

Structural and airborne sound propagation during the usage of everyday products in mobility, housing and work can induce discomfort and deteriorate health in the long-run (Basner et al., 2014). Therefore, great efforts are made to reduce sound pressure levels during product design. This is typically a two-stage process: (1) First, the sound characteristics of a design need to be evaluated. For this, discretization methods such as the finite element method (Zienkiewicz et al., 2005; Bathe, 2007) are applied on mechanical design models and the systems are numerically solved. This method yields a field solution over the design's spatial domain. Field solutions provide a physical quantity such as the vibration velocity at each point of the design geometry and each queried frequency. To obtain a compact description of the design, the field solutions are spatially averaged and converted to dB-scale,

resulting in the *frequency response*. (2) Second, the design needs to be changed to reduce the emitted sound. In general, it is desirable to keep the frequency response low, especially in frequency bands humans are sensitive to or where acoustic coupling is expected. Experienced engineers use frequency responses to directly devise sound reduction measures from experience, e.g. stiffening a design at a certain location to shift resonance frequencies.

The exploration of noise-reducing designs is limited by prohibitive simulation costs. Data-driven surrogate modelling is a promising technique that could circumvent this constraint by accelerating the evaluation of design candidates by several magnitudes. To assess the quality of such surrogate models and foster the development of well-performing models, structured benchmarks are required. While benchmarks exist for similar problems, such as directly predicting the solution of time-domain partial differential equations (Takamoto et al., 2022; Otness et al., 2021) and computational fluid dynamics (Takamoto et al., 2022; Bonnet et al., 2022), there is currently no structured benchmark for predicting responses in the frequency domain based on variable geometries. The only dataset dealing with frequency domain data we are aware of pertains to the field of electromagnetic compatibility (Schierholz et al., 2021). To address this gap, we consider vibrating plates excited by a harmonic force as a representative acoustic design problem and introduce a benchmark: Given a variation of a plate geometry and material properties, the goal is to predict the corresponding frequency response function. We vary scalar properties of the plate as well as the geometry by adding *beading patterns*, that change the acoustic properties of the plate (e.g. by shifting resonances) (Rothe, 2022). Plates are common in technical systems as they often function as a building block for more complex designs, e.g. in car bodies, lightweight walls or aircraft fuselages. Also, the characteristics of frequency responses of more complex systems do not systematically differ (Römer et al., 2021; Blech et al., 2021). From a machine learning perspective the problem is intriguing because simple input patterns cause complex, multi-peak frequency responses while adding beadings has a stiffening effect and reduces the number of resonance peaks in the investigated frequency range, resulting in simpler frequency responses. Also, this setup is different to existing benchmarks in that we do not look at the system's evolution over time but the response in the frequency domain.

To tackle frequency response prediction, we propose a novel operator model, named Frequency Query Operator (FQ-Operator). This model is trained to map plate geometries into the space of frequency response functions. In this context, operator means that the frequency response function can be evaluated at any frequency, making it infinitely dimensional rather than being limited to a fixed-size vector (Lu et al., 2019). This approach is closely related to implicit models for shape representation (e.g. Mescheder et al., 2018; Saito et al., 2019; Yu et al., 2020). Here, geometry is represented by a neural network, which can be queried at arbitrary points and predicts whether these points are inside or outside the surface. A challenge in frequency response prediction is to accurately predict resonance peaks. To address this, we combine approaches from the research areas of operator learning and implicit models.

Summarizing, our contributions are as follows: ① The novel benchmark dataset addressing frequency response prediction of a vibrating thin plates with varied geometries and stiffening patterns. As part of the benchmark, we propose three complementary metrics to quantify frequency response prediction quality. ② We evaluate existing methods on this dataset and report their scores for reference. These methods involve DeepONet (Lu et al., 2019) and Fourier Neural Operators (Li et al., 2020) among others. ③ We propose the query-based operator learning architecture, FQ-Operator, that outperforms existing methods on our vibrating-plates dataset.

## 2   RELATED WORK

**Acoustics.**   While research on surrogate models for the spatio-temporal evolution of vector fields is fairly common, directly predicting frequency responses through neural networks is an understudied problem. A general CNN architecture is applied in (Lanning et al., 2022) to calibrate the parameters of an analytical model for a composite column on a shake table. The data includes spectrograms representing the structural response in time-frequency domain. The frequency-domain response of acoustic metamaterials is considered in a material design task by conditional generative adversarial networks or reinforcement learning (Gurbuz et al., 2021; Shah et al., 2021; Lai et al., 2021). The frequency response of a multi-mass oscillator is predicted with transformer-based methods Schultz et al. (2023). Within the context of aeroacoustics, the propagation of a two-dimensional acoustic wave

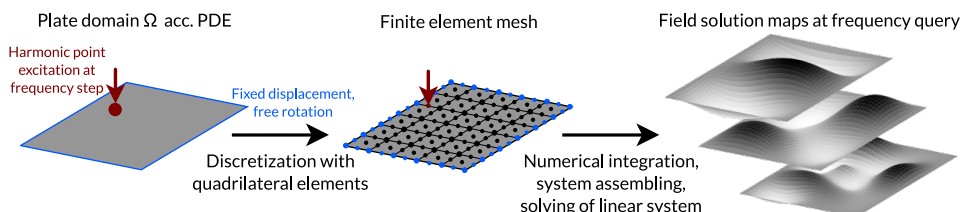

Figure 2: Process of the finite element solution in frequency domain in order to yield the field solutions at each frequency query.

while considering sound-scattering obstacles is predicted in time-domain by a CNN Alguacil et al. (2021; 2022). A review of machine learning in acoustics is given by (Bianco et al., 2019). Concerning benchmarks in acoustics, with (Hornikx et al., 2015), several acoustic benchmarks for numerical methods are available. However, these benchmarks do not systematically vary input geometries, making them not directly applicable to data-driven models.

**Scientific machine learning.** Data-driven machine learning techniques were successfully applied in many different disciplines within engineering and applied science; for example for alloy discovery (Rao et al., 2022), crystal structure prediction (Ryan et al., 2018), climate modeling (Rasp et al., 2018) and protein folding (Jumper et al., 2021). A popular use case for data-driven methods is to accelerate fluid dynamics, governed by the Navier-Stokes equations (Brunton et al., 2020; Kochkov et al., 2021; Obiols-Sales et al., 2020; Wang et al., 2019; Tompson et al., 2016).

The question of how to structure and train neural networks for predicting the solution of partial differential equations (PDE) has been the topic of intense research. Many methods investigate the inclusion of physics informed loss terms (Raissi et al., 2019; Haghighat et al., 2021; Krishnapriyan et al., 2021; Wang et al., 2019; Heilenkötter & Freudenberg, 2023). Some methods directly solve PDEs with neural networks as a surrogate model (Yu et al., 2018; Bu & Karpatne, 2021). Graph neural networks are often employed, e.g. for interaction of rigid (Battaglia et al., 2016; Sanchez-Gonzalez et al., 2020) and deformable (Battaglia et al., 2016; Sanchez-Gonzalez et al., 2020) objects as well as fluids (Sanchez-Gonzalez et al., 2020).

**Operator learning and implicit models.** A promising avenue of research for incorporating inductive biases for physical models has been operator learning (Lu et al., 2019; Li et al., 2020; Lu et al., 2022; Seidman et al., 2022; Kovachki et al., 2023). Operator learning structures neural networks such that instead of directly mapping from discrete input to output space, the neural network produces a function that can be evaluated at real values instead of a discrete grid. DeepONet (Lu et al., 2019) implements operator learning by taking the value at which it is evaluated as an input and processes this value in a separate branch. Fourier Neural Operators (Li et al., 2020) use a point-wise mapping to a latent space which is processed through a sequence of individual layers in Fourier space before being projected to the output space.

Implicit models (or coordinate-based representation) are models where location is utilized as an input to obtain a location-specific prediction, instead of predicting the entire grid at once and thus fit in the operator learning paradigm. Such models were used to represent shapes (Mescheder et al., 2018; Chen & Zhang, 2018; Park et al., 2019; Saito et al., 2019), later their representations were improved (Sitzmann et al., 2020; Tancik et al., 2020) and adapted for representing neural radiance fields (NeRFs) (Mildenhall et al., 2020; Yu et al., 2020). Our method applies techniques from these implicit models to operator learning.

## 3 DATASET AND BENCHMARK CONSTRUCTION

### 3.1 PROBLEM DEFINITION

Vibrating plates are common components in housings, walls and outer skins. In this work, we consider the vibration of a simply supported aluminum plate excited by a point force as a representative problem from this domain. "simply supported" means points on the edges cannot move up or down but can rotate. The rest of the plate can move freely. The mechanical problem is modeled by the following

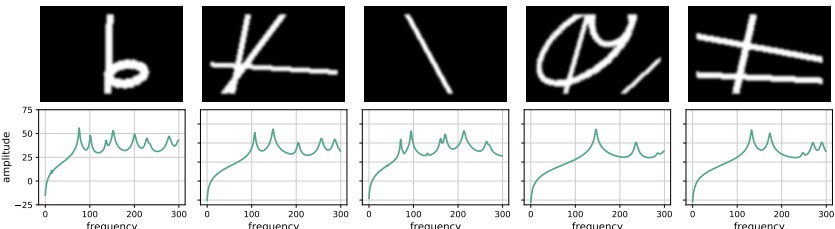

**Setting V-5000: Variable number of elements** (1 - 3 lines, 0 - 2 ellipses)

Figure 3: Example discretized plate geometries with corresponding frequency responses. Note, an intriguing property of this dataset is that the simpler the input, the more complex the output gets.

partial differential equation based on the plate theory by Mindlin (Mindlin, 1951):

$$B \, \nabla^4 u_z - \omega^2 \rho_s h \, u_z + \omega^2 \left( \frac{B\rho_s}{G} + \rho_s I \right) \, \nabla^2 u_z + \omega^4 I \frac{\rho_s^2}{G} \, u_z = p_l \tag{1}$$

In the equation, $u_z$ denotes the normal displacement of the plate structure as degree of freedom of interest. $B$ represents the bending stiffness, $\rho_s$ the density, $h$ the thickness, $G$ the shear modulus and $I$ the moment of inertia. The right hand-side excitation $p_l$ describes an applied pressure load. To obtain the reference solution (ground truth) of this mechanical problem, we use a numerical discretization technique, i.e. the finite element method (FEM). Plate geometries are represented by a discretization with finite domains (elements), which connect the entire domain and approximate the shape and the solution of the physical quantities by polynomial ansatz functions. The elements follow typical geometrical forms such as quadrilaterals, which are applied here for meshing the spatial dimensions of the plate as shown in Figure 2, center. As the last step in Figure 2, the discretized domain $\Omega$ is integrated in order to assemble the system of equations in the following form:

$$\left(\mathbf{K} - \omega^2 \mathbf{M}\right) \mathbf{y} = \mathbf{b} \tag{2}$$

The matrices $\mathbf{K}$ and $\mathbf{M}$ are stiffness and mass matrix, respectively, while $\mathbf{y}$ contains the degrees of freedom (field values such as displacement) and $\mathbf{b}$ the excitation forces. The linear system is solved by a parallel direct solver in order to receive the harmonic field solutions ($\mathbf{y}$) at the queried frequency steps. The solution vector contains the translational degree of freedom representing the normal vibration amplitude at the queried frequency at all discretization points within $\Omega$. The FEM converges to the exact solution by refining the mesh resolution. For details, see e.g. Atalla & Sgard (2015). We provide further details on the mechanical model and FEM setup in Appendix A.1. In general, the procedure of considering a differential equation for the mechanical domain, discretizing by FEM and solving the yielded sparse system above is similar for other acoustical systems.

A harmonic unit point load is applied at a fixed relative position near one corner of the plate. This way, all typical dynamic characteristics of the plate are excited, which is comparable to a realistic loading, e.g. by engines. Due to the non-central force location, the plate's response is not symmetric. Different load positions and distributed loads slightly change the dynamic response, but the characteristics remain. Point forces occur in real world applications, when a vibration source is coupled with the plate by a point connection (e.g. a plate being attached to a motor or compressor by a screw). Therefore, a point force excitation is used in many vibroacoustic settings (e.g. Yairi et al., 2002; Shorter & Langley, 2005). As we consider damping in the system, the amplitudes are finite within the plate resonances. The unit point load is applied in a frequency range of 1 to 300 Hz.

### 3.2 GEOMETRY VARIATION

The benchmark dataset is constructed to enable surrogate modeling and at the same time cover typical variations an engineer has to consider. To achieve this, we introduce two axes of variation: (1) beading patterns, imposed on the plate geometry, and (2) the geometry and material of the plate itself, concretely the width, length and thickness as well as the damping loss factor of the plate. These parameters along with fixed parameters as well as plots showing the effect of a damping and thickness variation are given in Appendix A.2.

**Beading patterns.** Beading patterns are stamped into the plate, which causes a stiffening effect and thus directly influences the field solutions and the frequency response function (Rothe, 2022). Adding

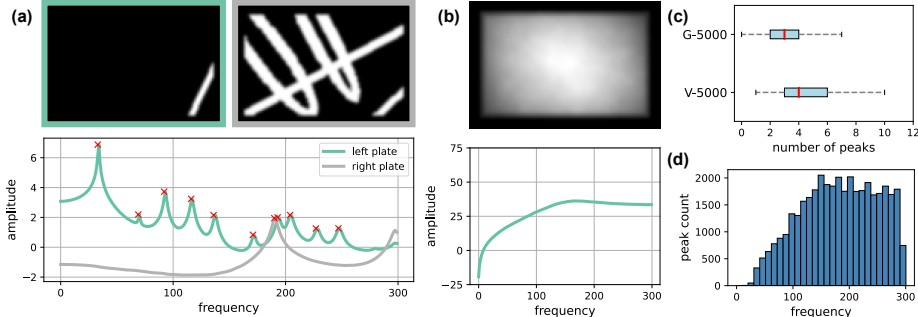

Figure 4: Dataset analysis. (a) shows two discretized plate geometries with their corresponding frequency response, the red crosses mark the detected peaks. (b) shows the mean plate design and frequency response. (c) shows number of peaks in different dataset settings. (d) shows the distribution of the peaks over the frequencies.

beadings to plates to make them more stiff is an established method (see for example corrugations of tin cans). We consider beadings composed of lines and ellipses. The position, orientation, width and size of the beadings are variable, inducing a large design space of possible beading patterns. The resulting beading patterns are represented as a height map $\mathbf{h} \in \mathbb{R}^{H \times W}$, where $H$ and $W$ are equal to the number of nodes in the FEM discretization (see Figure 2, center). They can directly be incorporated in the FEM mesh by shifting the z coordinates of the mesh according to the height map. The beading height is set to 20 mm and a Gaussian filter with standard deviation $\sigma = 1$ is applied to the beading pattern to ensure transitions at the beading edges are smooth and manufacturable. To actually produce the plates described in our work, stamps for the different beading patterns would need be manufactured.

**Dataset settings.** We construct two dataset splits: For the V-5000 setting, we fix the scalar geometry and material parameters and only impose a randomly sampled beading pattern on the plates. Specifically, $1 - 3$ lines and 0 - 2 ellipses are placed. Also, the width of the beading-elements is randomly sampled. Example plates are shown in Figure 3 along with their frequency responses. For the G-5000 setting, we apply the same beading pattern variation and additionally vary the plate geometry (length, width and thickness) as well as one material parameter (damping loss factor). For each setting, 5000 training and validation samples and 1000 test samples are generated.

**Dataset analysis.** The mean plate design shows a close to uniform distribution, with an area left free from beadings at the plate's edge (see Figure 4b). We find that the number of peaks corresponds with the beaded area: the greater the proportion of beaded mesh elements in a given plate, the fewer the peaks (see Figure 4a). This is due to additional beadings stiffening the plates, and it represents an interesting trait specific to our problem. The density of peaks is related to the frequency. As the frequency increases, so does the peak density. Starting from 150 Hz the peak density plateaus (see Figure 4d). The average number of peaks in the G-5000 setting is slightly smaller than in the V-5000 setting. This is influenced by the on average smaller plates being stiffer and therefore having less peaks in the frequency range (see Figure 4c).

### 3.3 EVALUATION METRICS

We propose three complementary metrics to measure the quality of the frequency response predictions.

**Mean squared error.** The *mean squared error (MSE)* is a well-known regression error measure: For the global deviation we compare the predicted $\hat{\mathbf{r}}$ and ground truth (obtained through simulation) frequency response $\mathbf{r}$ by the MSE error $\mathcal{E}_{\text{MSE}} = \sum_i (\hat{\mathbf{r}}_i - \mathbf{r}_i)^2$.

**Earth mover distance.** The *earth mover distance* (Pele & Werman, 2009; Rubner et al., 2000) expresses the work needed to transmute a distribution $P$ into another distribution $Q$. As a first step, the optimal flow $\hat{\gamma}$ is identified. Based on $\hat{\gamma}$ the earth mover distance is expressed as follows:

$$\mathcal{E}_{\text{EMD}}(P, Q) = \frac{\sum_{i,j} \hat{\gamma}_{ij} \cdot d_{ij}}{\sum_{i,j} \hat{\gamma}_{ij}} \quad \text{with } \hat{\gamma} = \min_{\gamma} \sum_{i,j} \gamma_{ij} \cdot d_{ij} \tag{3}$$

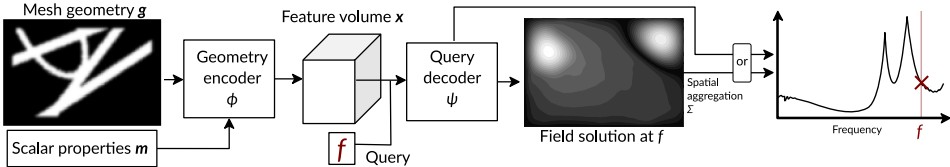

Figure 5: FQ-Operator method. The geometry encoder takes the mesh geometry and the scalar properties as input. The resulting feature volume along with a frequency query is passed to the query decoder, that either predicts a field solution or directly a frequency response. The field solution is aggregated to arrive at the frequency response at the query frequency f.

where $d_{ij}$ is the distance between bins $i$ and $j$ in $P$ and $Q$. Correspondingly, $\gamma_{ij}$ is the flow between bins i and j. We calculate the $\mathcal{E}_{\text{EMD}}$ based on the original amplitudes in $m/s$ that have not been transformed to the log-scale (dB) and normalize these amplitudes with the sum over all frequencies. As a consequence and unlike the MSE, $\mathcal{E}_{\text{EMD}}$ is invariant to the mean amplitude and only considers the shape of the frequency response. In this form, our metric is equivalent to the $W_1$ Wasserstein metric (Vaserstein, 1969; Cuturi, 2013).

**Peak frequency error.** To specifically address the prediction of frequency peaks, which are particularly relevant for engineers, we introduce a third metric called *peak frequency error*. The metric answers two questions: (1) Does the predicted frequency response contain the same peaks as the true response? (2) How far are corresponding ground truth and prediction peaks shifted against each other? To this end, we set up an algorithm that starts by detecting a set of peaks $P_{\text{GT}}$ in the ground truth and a set of peaks $P_{\text{PRED}}$ in the prediction using the `find_peaks` function in scipy (Virtanen et al., 2020) (examples in Appendix B). Then, we match these peaks pairwise using the Hungarian algorithm (Kuhn, 1955) based on the frequency distance. This allows us to determine the ratio between predicted and actual peaks $R_{peaks} = \frac{|P_{\text{PRED}}|}{|P_{\text{GT}}|}$. To provide a notion of the distribution, we report $D_{[0.25,0.75]}$, the 25 % and the 75 % quantile of $R_{peaks}$. We further report $\mathcal{E}_{\text{F}}$, the mean frequency distance of the matched peaks in Hz. These metrics enable straightforward interpretation of the results.

### 3.4 FREQUENCY RESPONSE GROUND TRUTH

To calculate the aggregate frequency response we take the spatial average of the field solutions, specifically the squared absolute velocity in z-direction (orthogonal to the plate). The squared absolute velocity is an important quantity in acoustics, since it is directly related to the radiated sound power of the plate (Sung & Jan, 1997). The result is then converted to the dB-scale by rescaling and taking the logarithm. To address numerical issues as well as facilitate an easier interpretation of the evaluation metrics, we normalize the frequency response and the field solution. To do this, we first take the log of the field solutions, to align it with the dB-scale of the frequency response. Then, we subtract the mean per frequency over all samples (depicted in Figure 4b for frequency response) and then divide by the overall standard deviation across all frequencies and samples. Small changes in the beading pattern can cause frequency shifts, potentially pushing peaks out of the considered frequency band. To reduce the effect of such edge cases, we predict frequency responses between 1 and 300 Hz but evaluate on the frequency band between 1 and 250 Hz.

## 4 FREQUENCY RESPONSE PREDICTION MODEL

Our goal is to predict a frequency response function $\mathcal{F}(\mathbf{g}, \mathbf{m})$, where $\mathcal{F}(\cdot)$ denotes an operator, mapping the mesh geometry $\mathbf{g}$ and a set of scalar parameters for geometry and material $\mathbf{m}$ to the frequency response function (Figure 5). Therefore, following the operator learning paradigm (Lu et al., 2019), $\mathcal{F}(\mathbf{g}, \mathbf{m})$ is defined for any, even non-integer, frequency $f$. In contrast, a grid-based vector output would only be defined at certain frequencies. We divide this problem into processing geometry input by an encoder $\Phi$ and evaluating the output function using a decoder $\Psi$:

$$\mathcal{F}(\mathbf{g}, \mathbf{m})(f) = \Psi(\Phi(\mathbf{g}, \mathbf{m}), f) \tag{4}$$

This formulation is not only common in operator learning but shares similarities with implicit models, for instance by Saito et al. (2019) in the context of 3d shape prediction. For frequency response prediction, we hypothesize that a key challenge lies in a precise prediction of resonance peaks for which an implicit formulation is better suited than using a fixed grid.

## 4.1 GEOMETRY ENCODER Φ

To parse the plate geometry into a a feature vector, we employ three variants: ResNet18 (He et al., 2016, RN18), the vision transformer (Dosovitskiy et al., 2020; Vaswani et al., 2017, ViT) and the encoder part of a UNet (Ronneberger et al., 2015). Processing the geometry mesh with CNNs for 2d images is possible, because the 3d mesh can be represented in 2d as depth over a planar grid structure. For the RN18, we replace batch normalization with layer normalization (Ba et al., 2016), as we found this to work substantially better. Compared to the CNN-based RN18, the ViT architecture supports interactions across different image regions in early layers. For both, the RN18 and the ViT encoder, we obtain a feature vector $\mathbf{x}$ by average pooling the last feature map. Since the UNet generates field solutions, no pooling is applied.

**FiLM conditioning.** For including the scalar geometry information and the loss factor, we introduce a film layer (Perez et al., 2018). The film layer first encodes the scalar parameters with a linear layer. The resulting encoding is then multiplied element-wise with the feature of the encoder and a bias is added. This operation is applied before the last layer of the geometry encoder (UNet) or after it (RN18, ViT).

## 4.2 FREQUENCY QUERY OPERATOR DECODER Ψ

**FQO-RN18 and FQO-ViT - MLP-based decoder.** Having obtained an encoding of the plate geometry and properties $\mathbf{x}$, a decoder now takes this as well as a frequency query as input and maps them towards a prediction. For the RN18 and ViT geometry encoders, the decoder is implemented by an MLP $g$ taking both $\mathbf{x}$ and a scalar frequency value $f$ as input to predict the response for that specific query frequency, i.e. $g(\mathbf{x}, f) \in \mathbb{R}$. By querying the decoder with all frequencies individually, we obtain results for the frequency band between 1 and 300 Hz. The MLP has six hidden layers with 512 dimensions each and ReLU activations.

**FQO-UNet - field solution mediated decoder.** To incorporate physics-based contraints, we employ the UNet decoder to predict the field solutions instead of directly predicting the frequency response. A frequency query is appended to the channels of the feature vector produced by the UNet encoder and the network is trained to predict the field solution for this specific query. Note, the predicted field solution is converted to the frequency response by spatial averaging of the squared field quantity. The training loss is the (unweighted) average of the mean squared error of the normalized field solution and frequency response. The downside of this approach is that the decoder has to be evaluated 300 times for 300 frequency queries, slowing down inference and training and leading to higher memory demands. To interpolate between a query-based decoder and predicting all frequency response values at once, we can predict a differently sized neighborhood around the frequency query. Reducing the necessary computations by a factor of five, we opt to map to five frequencies per query. This choice is ablated in Appendix D. The runtime of one forward pass of our methods is reported in Appendix C.6.

## 4.3 BASELINES

To compare the FQ-Operator method we further report baseline results on the following alternative methods: A $k$-Nearest Neighbors regressor, that finds the nearest neighbors in the latent space of an autoencoder. Two grid-based prediction methods that share the FQO-RN18 and FQO-UNet architectures, but do not employ frequency queries. They instead predict frequency responses or the field solution for all 300 frequencies at once. DeepONet (Lu et al., 2019), with a RN18 as trunk net and a MLP to encode the query frequencies as a branch net. Two architectures based on Fourier Neural Operators (Li et al., 2020). One employing a FNO as a replacement for the query-based decoder based on RN18 features. The second directly takes the input geometry and is trained to map it to the field solutions. See Appendix C for details on all architectures and specifics on the training procedure.

## 5 EXPERIMENTS

We train the FQ-Operator variations and baseline methods on the vibrating-plates dataset (see Table 1). We find that (a) the FQ-Operator variations outperforms grid-based baselines as well as other operator learning methods and (b) predicting the field solutions and then transforming them to the frequency response leads to better results than directly predicting the frequency response. Regarding (1), we find that the FQ-Operator variations consistently yield better predictions than equivalent grid-based

| | FS | V-5000 | | | | G-5000 | | | |
|---|---|---|---|---|---|---|---|---|---|
| | | $\mathcal{E}_{\mathrm{MSE}}$ | $\mathcal{E}_{\mathrm{EMD}}$ | $D_{[0.25,0.75]}$ | $\mathcal{E}_{\mathrm{F}}$ | $\mathcal{E}_{\mathrm{MSE}}$ | $\mathcal{E}_{\mathrm{EMD}}$ | $D_{[0.25,0.75]}$ | $\mathcal{E}_{\mathrm{F}}$ |
| $k$-NN | - | 0.42 | 19.78 | [0.33, 0.67] | 11.5 | 0.69 | 28.87 | [0.00, 0.50] | 20.9 |
| RN18 + FNO | - | 0.18 | 8.70 | [0.50, 1.00] | 4.4 | 0.25 | 13.33 | [0.50, 1.00] | 7.4 |
| DeepONet | - | 0.25 | 14.04 | [0.43, 0.67] | 5.0 | 0.34 | 19.57 | [0.20, 0.50] | 9.6 |
| FNO (field solution) | ✓ | 0.26 | 13.70 | [0.50, 1.00] | 6.9 | 0.24 | 13.25 | [0.50, 1.00] | 7.1 |
| Grid-RN18 | - | 0.23 | 10.60 | [0.50, 1.00] | 4.7 | 0.22 | 11.14 | [0.50, 1.00] | 6.1 |
| Grid-UNet | ✓ | 0.12 | 8.13 | [0.67, 1.00] | 3.1 | 0.15 | 9.22 | [0.50, 1.00] | 5.2 |
| FQO-ViT | - | 0.29 | 13.81 | [0.50, 0.75] | 6.1 | 0.30 | 14.32 | [0.33, 1.00] | 8.3 |
| FQO-RN18 | - | 0.17 | 8.58 | [0.60, 1.00] | 3.9 | 0.19 | 10.04 | [0.53, 1.00] | 5.6 |
| FQO-UNet | ✓ | **0.09** | **5.99** | **[0.74**, 1.00] | **2.7** | **0.13** | **7.14** | **[0.67**, 1.00] | **4.6** |

Table 1: Test results for frequency response prediction. FS indicates if field solutions are predicted from which the frequency response is derived. We report mean and standard deviation results for multiple runs of the FQO-UNet in Appendix E.4.

methods, where responses for all frequencies are predicted at once: The $\mathcal{E}_{\mathrm{MSE}}$ and the $\mathcal{E}_{\mathrm{EMD}}$ are lower, more peaks are reproduced and the peak positions are more precise. Regarding (2), FQO-UNet strongly outperforms the FQO-RN18, that directly predict frequency responses, which we attribute to the richer training data of field solutions.

Despite using the same RN18 geometry encoder as FQO-RN18, DeepONet (Lu et al., 2019) performs worse. We assume that this is due to its approach of incorporating frequency information through a single weighted summation, which limits the model's expressivity (Seidman et al., 2022). In contrast, FQ-Operator introduces the queried frequency earlier into the model. We also test two Fourier Neural Operator (Li et al., 2020, FNO) baselines: the first, RN18 + FNO, which substitutes the query-based decoder with an FNO decoder, slightly underperforms compared to FQO-RN18 on both datasets. The second FNO baseline, trained directly to predict field solutions, yields poorer results despite having access to richer training data. Ablations on several architecture choices are provided in Appendix D. The G-5000 setting seems to yield slightly worse results than the V-5000 setting but the differences seem minor. The small difference is surprising because the space of plates in the G-5000 setting is a superset of the V-5000 space. One reason for this might be the average number of peaks in the frequency response: the plates in G-5000 are on average smaller and because of this stiffer, leading to less peaks (on average 2.5 in G-5000 vs. 3.5 in V-5000 ). This interpretation is supported by the fact that the average error becomes higher with increasing frequency (Figure 6f) and thus increasing peak density (Figure 4d).

Looking at a prediction example (Figure 6a-d), we see that the predicted field solution from the FQO-UNet has some differences to the ground truth. The prediction captures the three modes and their position quite well, but the shape of the modes is less regular than in the field solution. Despite that, the resulting frequency response prediction at $f = 149$ is close to the FEM reference (ground truth). In comparison to the grid-based prediction, where peaks tend to be blurry, the frequency response peaks generated by FQO-UNet are more pronounced. Additional video form visualizations are provided in the supplementary material.

**Transfer learning.** To quantify to which degree features learned on a subset of the design space transfer to a different subset, we split the V-5000 setting into two equally-sized parts based on the number of mesh elements that are part of a beading. The "more beadings" set contains only 2.5 peaks on average because the plates are stiffened by the beadings, compared to 4.4 peaks on average for the "less beadings" set. We find that the training on plates with less beadings leads to a smaller drop in prediction quality (see Table 2). This indicates that training on data with more complex frequency responses might be more efficient. We further report results on evaluating our method on frequencies not seen during training in Appendix E.3.

**Sampling efficiency.** We train the FQO-UNet and the FQO-RN18 with reduced numbers of samples (see Figure 6e). Note, that the FQO-UNet with a quarter of the training data has close to the same prediction quality as the FQO-RN18 with full training data. This highlights the benefit of including the field solutions into the training process. Quantitative results are given in Appendix E.2 for both dataset settings.

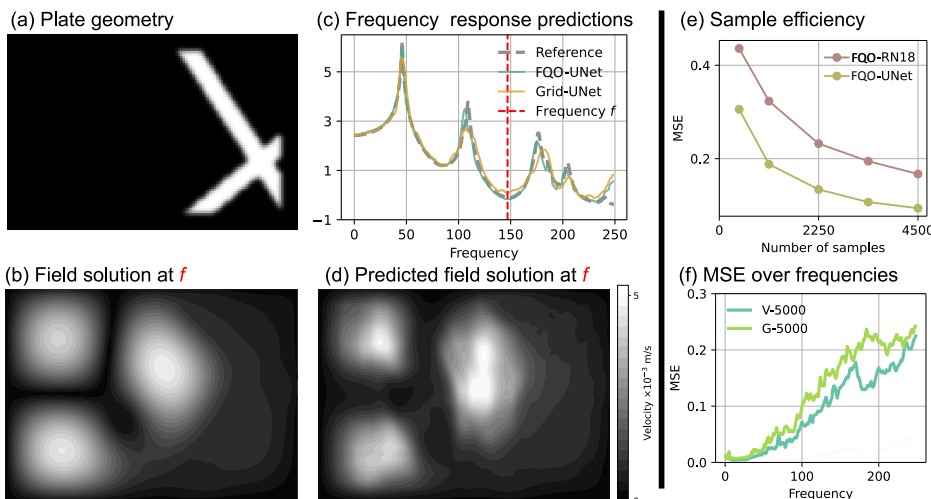

Figure 6: Results. (b) to (d) show the field solution at one frequency and prediction for the plate geometry in (a) from FQO-UNet. (e) shows the test MSE for training two methods with reduced numbers of samples from V-5000. (f) shows the MSE for the FQO-UNet for different frequencies.

| | less beadings $\mapsto$ more beadings | | | | more beadings $\mapsto$ less beadings | | | |
|---|---|---|---|---|---|---|---|---|
| | $\mathcal{E}_{\text{MSE}}$ | $\mathcal{E}_{\text{EMD}}$ | $D_{[0.25, 0.75]}$ | $\mathcal{E}_{\text{F}}$ | $\mathcal{E}_{\text{MSE}}$ | $\mathcal{E}_{\text{EMD}}$ | $D_{[0.25, 0.75]}$ | $\mathcal{E}_{\text{F}}$ |
| FQO-RN18 (origin) | 0.25 | 11.59 | [0.50, 0.75] | 4.6 | 0.21 | 10.01 | [0.67, 1.00] | 5.2 |
| FQO-RN18 | 0.32 | 12.68 | [0.67, 1.00] | 8.4 | 0.45 | 19.37 | [0.33, 0.60] | 6.2 |
| FQO-UNet (origin) | 0.15 | 8.67 | [0.67, 1.00] | 3.6 | 0.12 | 7.28 | [0.67, 1.00] | 3.3 |
| FQO-UNet | 0.17 | 9.33 | [0.67, 1.00] | 5.2 | 0.33 | 15.66 | [0.40, 0.75] | 5.1 |

Table 2: Transfer learning performance: We split V-5000 into two halves based on amount of beadings and evaluate transfer learning performance across these splits: training subset $\mapsto$ test subset. The gray rows denote validation results on the original subset that has been used for training. We provide baseline method results for this experiment in Appendix E.1.

## 6 CONCLUSION

We introduce the novel problem of data-driven vibroacoustic frequency response prediction from variable geometries. To this end, we created the vibrating-plates dataset with an associated benchmark to foster the development of new methods and provide reference scores with respect to related methods, for transfer learning and sample efficiency. We propose the FQ-Operator method to address frequency response prediction and show that our model compares favorably against the DeepONet and FNO baselines. In general, differentiable surrogate models show great potential for speeding-up frequency response prediction over finite elements method: Our models achieved speed-up factors of around 4 to 6 orders of magnitude. In terms of accuracy, our experiments demonstrate that such tasks can be learned with data-driven methods but specific architectures are necessary. Key findings of this work are that query-based approaches consistently outperform grid-based approaches and predicting the frequency response mediated by field solutions improves quality over direct prediction. We expect our benchmark to foster the discovery of inductive biases for frequency domain prediction, potentially involving components like different positional encodings, attention mechanisms, recurrence and graph neural networks.

**Limitations.** Our dataset and method serve as an initial step in the development of surrogate models for acoustic frequency response prediction. Our dataset focuses on plates, a common geometric primitive used in a great number of applications. However, many structures beyond plates exist, involving curved surfaces, multi-component geometries and complex material parameters. While FQ-Operator remains applicable in such cases, appropriate encoder and decoders would need to be designed. As more complex geometries will incur computational costs of FEM simulations, a key question will be how to enhance sample-efficiency even further.

**Ethics statement.** Noise reduction holds significance not just for the health of passengers in vehicles and urban residents but also for broadening acceptance of wind turbines and heat pumps. Findings in this work could be applicable to more complex problems and other engineering disciplines. Surrogate models can introduce a new source of error into the simulation process, especially when departing far from the training data. Users need to be aware of both the reliability and limitations of surrogate models.

**Reproducibility statement.** During the review process, the dataset and the code repository can ae accessed via https://anonymous.4open.science/r/FRONet-5536. Hyperparameters and architecture details are reported in Appendix C. Complete details for evaluating or training any architecture described in this work are available in the code repository. All models can be trained on a single GPU. For publication, the dataset will be uploaded to a dataset repository hosted by a university and is licensed under CC-BY-NC 4.0.

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

# A   DATASET CONSTRUCTION

## A.1   THE MECHANICAL MODEL

The mechanical model serves as reference solution and is applied to train the network. For moderately thin plates, the plate theory by Mindlin is a valid differential equation (Mindlin, 1951):

$$B \, \nabla^4 u_z - \omega^2 \rho_s h \, u_z + \omega^2 \left( \frac{B \rho_s}{G} + \rho_s I \right) \, \nabla^2 u_z + \omega^4 I \frac{\rho_s^2}{G} \, u_z = p_l,$$

This equation is combined with a disc formulation for in-plane loads in order to receive a shell formulation for the mechanical description of arbitrarily formed moderately thin structures considering in-plane and transverse loads. The plate part is the dominating and important part for resolving bending waves. In the equation, $u_z$ denotes the normal displacement of the plate structure as degree of freedom of interest. $B$ represents the bending stiffness, $\rho_s$ the density, $h$ the thickness, $G$ the shear modulus and $I$ the moment of inertia. The right hand-side excitation $p_l$ describes an applied pressure load, which is converted to point forces through integration. The equation is transformed into an integral form by weighted residuals, discretized using finite elements and integrated numerically. In particular, we use shell elements with 9 nodes and quadratic ansatz functions. The integration delivers the sparse linear system of equations, which is depicted in the paper. This linear system is solved using the direct solver MUMPS (Amestoy et al., 2000) with a specialized FEM implementation for acoustics (Sreekumar & Langer, 2023). The discretization is chosen, such that the bending waves are resolved with a minimum of 10 nodes. The bending wave length $\lambda_B$ of a plate can be calculated by

$$\lambda_B = \sqrt{\frac{2\pi}{f}} \sqrt[4]{\frac{Et^2}{12(1 - \nu^2)\rho}}, \tag{5}$$

where $E$ is the Young's modulus, $t$ the thickness, $\nu$ the Poisson ratio and $\rho$ the density of the plate. The final discretization is set to $121 \times 81$, which is sufficient for convergence.

## A.2   DATASETS

Geometric and material parameters of the plate are summarized in Table 3. Computing one frequency sweep with 300 frequency steps for one sample plate geometry takes 2 minutes and 19 seconds on a workstation with a 2 Ghz CPU (20 physical cores).

| Geometry | | | Material (Aluminum) | | | |
|---|---|---|---|---|---|---|
| length | width | thickness | density | Young's mod. | Poisson ratio | loss factor |
| 0.9 m | 0.6 m | 0.003 m | 2700 kg/m$^3$ | 7e10 N/m$^2$ | 0.3 | 0.02 |

Table 3: Plate properties, including geometry and material parameters.

| length | width | thickness | loss factor |
|---|---|---|---|
| 0.6 - 0.9 m | 0.4 - 0.6 m | 0.002 - 0.004 m | 0.01 - 0.03 |

Table 4: Geometry and physical variation parameters for G-5000 dataset.

For the G-5000 dataset, we additionally vary the geometry and material parameters. Figure 7 demonstrates the effect of a variation in the thickness and the loss factor. For this plot a one-at-a time parameter variation is performed, thus all parameter values are at nominal value except the variation parameter. Table 4 gives the range of parameters. We uniformly sample from the given ranges. Both settings are compared in Table 5.

| Setting | Geom. | Sample Space | | | Sample Number | |
|---|---|---|---|---|---|---|
| | | Lines | Ellipses | Width | Train | Test |
| V-5000 | fix | 1 - 3 | 0 - 2 | 30 - 70 | 5000 | 1000 |
| G-5000 | vary | 1 - 3 | 0 - 2 | 40 - 60 | 5000 | 1000 |

Table 5: Dataset settings. Width is the width of lines and ellipses in mm. Geometry (geom.) involves plate size, thickness and material.

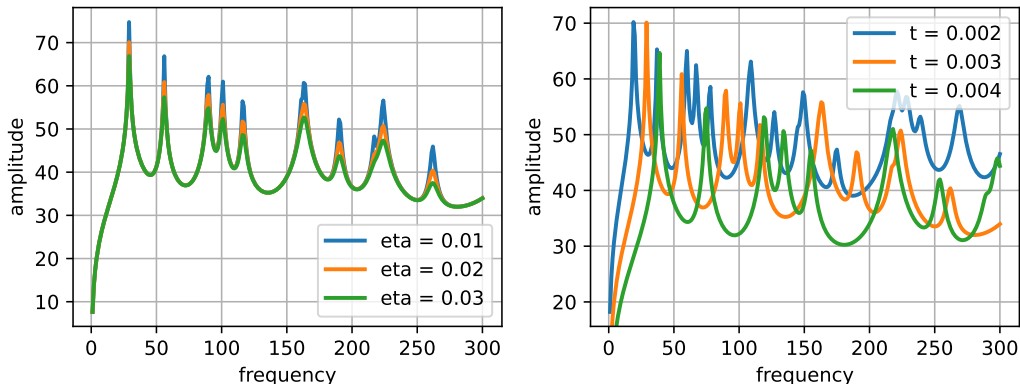

Figure 7: One-at-a-time parameter variation of the thickness parameter and the damping loss factor. Increasing the damping reduces the amplitudes at the resonance peaks. Increasing the plate thickness increases the stiffness of the plate and thus shifts the resonance peaks towards higher frequencies

## B METRICS - PEAK FREQUENCY ERROR

We provide examples of the find_peak operation which serves as the basis for the peak frequency error on ground truth (Fig. 8) and predictions (Fig. 9, using a kNN baseline) and visualize the matched peaks for calculating the peak frequency error (Fig. 10). Note that find_peaks is run with the prominence threshold set to 0.5 meaning that the peak must be at least 0.5 units higher than their surroundings.

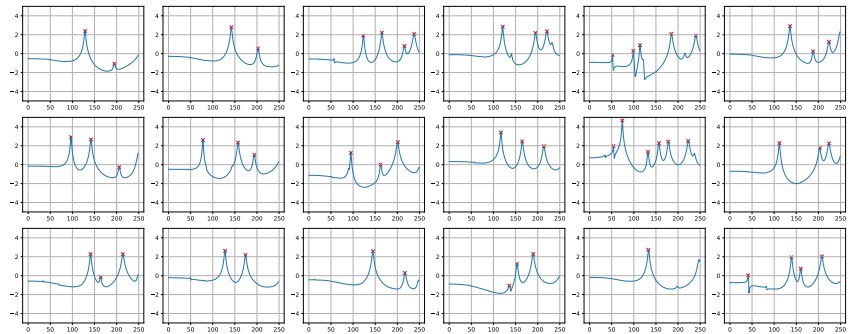

Figure 8: Find peak results on random ground truth samples.

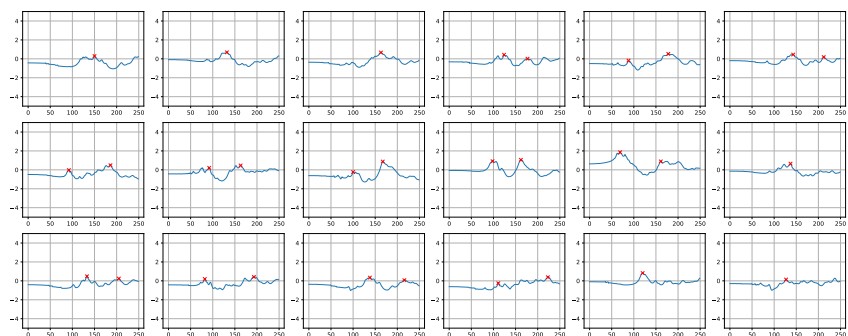

Figure 9: Find peak results on predictions.

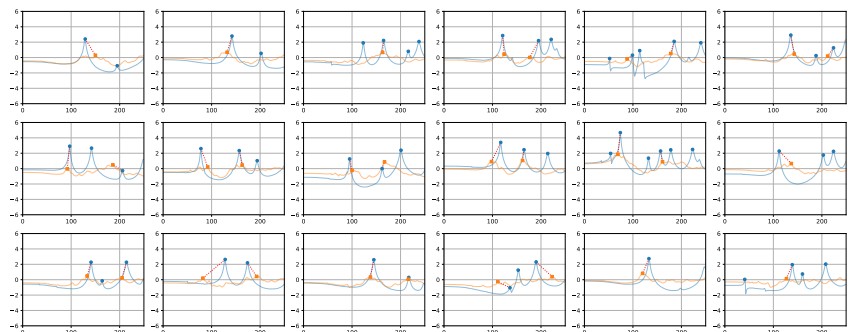

Figure 10: Visualization of the matching between ground truth (blue) and prediction (orange) peaks. Matched peaks are indicated in red.

## C  ARCHITECTURES

In the following, we give an overview of the employed neural architectures. For more details, we refer to the codebase. Table 6 gives an overview of the number of parameters for the different models.

### C.1  FQ-OPERATOR METHODS

Following our framework of partitioning the task of frequency response prediction into a geometry encoder and a query-based decoder, we test several different combinations of architectures for encoder and decoder.

| | # parameters in Mio |
|---|---|
| $k$-NN | None |
| RN18 + FNO | 11.4 |
| DeepONet | 11.3 |
| FNO (field solution) | 17.0 |
| Grid-RN18 | 12.9 |
| Grid-UNet | 24.0 |
| FQO-ViT | 6.9 |
| FQO-RN18 | 12.7 |
| FQO-UNet | 6.0 |

Table 6

**FQO-RN18 and FQO-ViT.** To directly predict the frequency response instead of predicting the field solutions and then transforming it to the frequency response, we try a ResNet He et al. (2016) and a vision transformer (ViT) Dosovitskiy et al. (2020) as geometry encoders. For the ResNet, we opt for the ResNet18 backbone. We replace batch normalization with layer normalization Ba et al. (2016), as we found this to work substantially better. In addition, we employ the visual transformer architecture Dosovitskiy et al. (2020). The ViT supports interactions across different image regions in early layers. We use a variation of the ViT-Base configuration with a reduced token size of 192, an intermediate size of 768 and three attention heads. For both the RN18 and the ViT encoder, we obtain the $d$-dimensional global feature $\mathbf{x}$ through average pooling from the last feature map or the encoded tokens. As a query-based decoder, we employ an MLP. The MLP $r$ takes both $\mathbf{x}$ and a scalar frequency value $f$ as input to predict the response for that specific query frequency, i.e. $r(\mathbf{x}, f) \in \mathbb{R}$. By querying the decoder with all frequencies individually, we obtain results for the frequency band between 1 and 300 Hz. The MLP has six hidden layers with 512 dimensions each and ReLU activations.

**FQO-UNet.** To predict the field solution and then transform it to the frequency response, we employ a UNet. To encode the geometry and scalar parameters we employ the encoder part of the U-Net. For the UNet decoder we then concatenate the query-frequency. The UNet consists of three contraction blocks, two spatial-shape-preserving blocks and two expansion blocks. The UNet is then trained to predict five field solutions in half the size of the input mesh.

## C.2 GRID-BASED METHODS

To provide a direct comparison to the query-based approach, two methods that predict all frequency responses at once are tested.

**Grid-RN18.** The same RN18 is used to generate a global feature $\mathbf{x}$ as in the FQO-RN18. Given $\mathbf{x}$, an MLP $r$ predicts the frequency response on a fixed 1 Hz interval grid, with $r(\mathbf{x}) \in \mathbb{R}^{300}$. We employ six hidden layers with 512 dimensions each and ReLU activations.

**Grid-based U-Net.** For the grid-based U-Net we also employ the same architecture as for the query-variation but double the number of channels to account for the larger number of predictions that the network has to produce at once. This yielded to comparatively smaller errors. The U-Net is trained to predict all 300 field solutions at once in half resolution in the same way as the query-variation.

## C.3 OTHER OPERATOR LEARNING METHODS

**RN18 + Fourier Neural Operator (FNO).** A 1d FNO as constructed by Li et al. (2020) takes as input $\mathbf{x}$ processed by a linear layer to size 300, the number of frequencies to be predicted. We keep 16 modes and use four FNO blocks as in the original implementation.

**DeepONet.** We further test a DeepONet with the RN18 as branch network and as trunk network, a four layer MLP of width 128 and 512 as output width to match the size of $\mathbf{x}$. The trunk network processes the frequency queries and is then combined with $\mathbf{x}$ to produce the prediction Lu et al. (2019; 2022). Note, the RN18 branch network is the same as the encoder of the FQO-RN18.

**FNO (field solution).** The FNO takes as input the beading pattern geometry interpolated to half the resolution. The 2d FNO then consists of four FNO blocks with 128 hidden channels and finally 300 output channels to map to the 300 field solutions in half the resolution. In the FNO blocks, 16 modes are preserved after the Fourier transform.

## C.4 K-NEAREST NEIGHBORS ($k$-NN)

We further test a $k$-Nearest Neighbors algorithm as a baseline, which predicts the frequency response of a plate as the mean frequency response of the $k$ closest plates in the training set. To determine the distance between different plate designs, we use the cosine distance on the 96-dimensional latent space of a convolutional autoencoder (Hinton & Salakhutdinov, 2006) trained on the beading pattern geometries. The normalized scalar properties are appended to the latent space to include them. To

obtain a prediction, the frequency responses of the $k$ neighbors are averaged and the optimal $k$ in the range $[1, 25]$ is empirically determined to minimize the MSE. The chosen $k$ is 14 and 22 for the V-5000 and the G-5000 setting respectively. Determining the nearest neighbors directly in the geometry space was tried out, but yielded worse results.

## C.5 TRAINING

The networks are trained using the AdamW optimizer (Loshchilov & Hutter, 2017). We further choose a cosine learning rate schedule with a warm-up period (Loshchilov & Hutter, 2016). The maximum learning rate is set to $0.001$, with the exception of the FQO-UNet, for which it is set to $0.0005$. In total, the networks are trained for 500 epochs. To validate, we define 500 samples from the training dataset as validation data and choose the checkpoint with the lowest MSE on these samples. We report evaluation results on the previously unseen test set.

## C.6 SPEED COMPARISON

| Model | Time (s) |
|---|---|
| RN18 + FNO | 0.006 |
| FQO-UNet | 0.763 |
| FQO-ViT | 0.008 |
| FQO-RN18 | 0.007 |
| DeepONet | 0.006 |
| Grid-UNet | 0.041 |
| Grid-RN18 | 0.006 |
| FNO (field solution) | 0.010 |

Table 7: Speed comparison for a forward pass for a batch of 32 plate geometries on a RTX A5000. In comparison, solving one geometry via FEM takes 2 minutes and 19 seconds. The slowest deep learning method is then around 5900 times faster,

# D ABLATIONS

To evaluate the number of frequency response predictions per query, we refer to Table 8.

For results in varying the depth and width of the mlp used in the query-based RN18 architecture, we refer to Table 9. By standard, a depth of 6 and a width of 512 is chosen.

For ablating the RN18 + FNO decoder architecture, we vary: The number of modes that are kept after the Fourier transform, the number of FNO blocks and the number of hidden channels. By standard, 16 modes are kept, 4 FNO blocks are applied, and 64 hidden channels are used. The results are detailed in Table 10.

| | **V-5000** | | | | |
|---|---|---|---|---|---|
| Number of predictions | $\mathcal{E}_{\text{MSE}}$ | $\mathcal{E}_{\text{EMD}}$ | $D_{[0.25,0.75]}$ | $\mathcal{E}_{\text{F}}$ | total runtime |
| 1 | 0.090 | 6.02 | [0.75, 1.00] | 2.7 | 7d 14h |
| 5 | 0.094 | 5.99 | [0.74, 1.00] | 2.7 | 1d 5h |
| 10 | 0.100 | 6.32 | [0.67, 1.00] | 2.8 | 18h |
| 30 | 0.118 | 7.27 | [0.67, 1.00] | 3.3 | 6h |

Table 8: Ablation results of number of frequency predictions per query on V-5000 dataset. Runtime on RTX A5000.

| | **V-5000** | | | |
|---|---|---|---|---|
| | $\mathcal{E}_{\text{MSE}}$ | $\mathcal{E}_{\text{EMD}}$ | $D_{[0.25,0.75]}$ | $\mathcal{E}_{\text{F}}$ |
| Depth 4 | 0.17 | 9.25 | [0.50, 1.00] | 3.3 |
| **Depth 6** | 0.16 | 10.04 | [0.50, 1.00] | 3.4 |
| Depth 8 | 0.17 | 9.20 | [0.60, 1.00] | 3.7 |
| Width 256 | 0.17 | 9.16 | [0.50, 1.00] | 3.4 |
| **Width 512** | 0.16 | 8.78 | [0.60, 1.00] | 3.8 |
| Width 1024 | 0.17 | 8.90 | [0.60, 1.00] | 3.8 |

Table 9: Ablation results of depth and width of query-based decoder with the FQO-RN18 on V-5000 dataset. The standard parameters for width and depth are 512 and 6 respectively.

| | **V-5000** | | | |
|---|---|---|---|---|
| | $\mathcal{E}_{\text{MSE}}$ | $\mathcal{E}_{\text{EMD}}$ | $D_{[0.25,0.75]}$ | $\mathcal{E}_{\text{F}}$ |
| **16 modes** | 0.18 | 9.05 | [0.50, 1.00] | 4.3 |
| 32 modes | 0.18 | 9.01 | [0.50, 1.00] | 4.0 |
| 64 modes | 0.18 | 8.50 | [0.60, 1.00] | 4.5 |
| 2 FNO blocks | 0.18 | 8.41 | [0.50, 1.00] | 4.1 |
| **4 FNO blocks** | 0.18 | 9.05 | [0.50, 1.00] | 4.3 |
| 8 FNO blocks | 0.19 | 9.38 | [0.50, 1.00] | 4.2 |
| 32 hidden channels | 0.18 | 9.69 | [0.50, 0.75] | 3.5 |
| **64 hidden channels** | 0.18 | 9.05 | [0.50, 1.00] | 4.3 |
| 128 hidden channels | 0.18 | 8.67 | [0.50, 1.00] | 4.5 |

Table 10: Ablation results for the number of modes that are kept after the Fourier transform, the number of FNO blocks and the number of hidden channels for the FNO decoder with the RN18 model. By standard, 16 modes are kept, 4 FNO blocks are applied, and 64 hidden channels are used.

| | V-5000 | | | |
| --- | --- | --- | --- | --- |
| | $\mathcal{E}_{\text{MSE}}$ | $\mathcal{E}_{\text{EMD}}$ | $D_{[0.25,0.75]}$ | $\mathcal{E}_{\text{F}}$ |
| **16 modes** | 0.28 | 15.20 | [0.50, 1.00] | 7.2 |
| 32 modes | 0.29 | 14.63 | [0.50, 0.80] | 6.8 |
| 64 hidden channels | 0.28 | 15.20 | [0.50, 1.00] | 7.2 |
| **128 hidden channels** | 0.26 | 13.85 | [0.50, 1.00] | 6.7 |

Table 11: Ablation results for FNO (fields solutions). Standard is 64 hidden channels and 16 models.

| | less beadings $\mapsto$ more beadings | | | | more beadings $\mapsto$ less beadings | | | |
| --- | --- | --- | --- | --- | --- | --- | --- | --- |
| | $\mathcal{E}_{\text{MSE}}$ | $\mathcal{E}_{\text{EMD}}$ | $D_{[0.25,0.75]}$ | $\mathcal{E}_{\text{F}}$ | $\mathcal{E}_{\text{MSE}}$ | $\mathcal{E}_{\text{EMD}}$ | $D_{[0.25,0.75]}$ | $\mathcal{E}_{\text{F}}$ |
| RN18+FNO (origin) | 0.26 | 13.75 | [0.40, 0.67] | 4.6 | 0.21 | 10.25 | [0.50, 1.00] | 5.2 |
| RN18+FNO | 0.29 | 14.33 | [0.50, 1.00] | 7.9 | 0.67 | 27.77 | [0.50, 0.72] | 8.2 |
| FNO (field solution, origin) | 0.34 | 16.15 | [0.40, 0.75] | 7.5 | 0.32 | 15.12 | [0.50, 1.00] | 9.3 |
| FNO (field solution) | 0.48 | 20.26 | [0.50, 1.00] | 13.6 | 1.02 | 26.76 | [0.32, 0.60] | 10.8 |
| Grid-RN18 (origin) | 0.31 | 13.79 | [0.40, 0.67] | 5.5 | 0.23 | 11.63 | [0.50, 1.00] | 5.2 |
| Grid-RN18 | 0.35 | 16.13 | [0.50, 1.00] | 10.0 | 0.62 | 19.42 | [0.50, 0.75] | 8.5 |
| Grid-UNet (origin) | 0.19 | 10.63 | [0.50, 0.80] | 4.0 | 0.13 | 8.24 | [0.67, 1.00] | 3.7 |
| Grid-UNet | 0.21 | 12.02 | [0.67, 1.00] | 6.5 | 0.56 | 21.41 | [0.60, 0.86] | 8.1 |

Table 12: Transfer learning performance: We split V-5000 into two halves based on amount of beadings and evaluate transfer learning performance across these splits: training subset $\mapsto$ test subset. The gray rows denote validation results on the original subset that has been used for training.

# E  ADDITIONAL RESULTS

## E.1  TRANSFER LEARNING

To provide baseline method results for the transfer learning setting with less beading or more beadings we refer to Table 12.

## E.2  SAMPLE EFFICIENCY

To provide full baseline results for the training with a reduced amount of samples, we refer to Table 13.

| | V-5000 | | | | G-5000 | | | |
| --- | --- | --- | --- | --- | --- | --- | --- | --- |
| | $\mathcal{E}_{\text{MSE}}$ | $\mathcal{E}_{\text{EMD}}$ | $D_{[0.25,0.75]}$ | $\mathcal{E}_{\text{F}}$ | $\mathcal{E}_{\text{MSE}}$ | $\mathcal{E}_{\text{EMD}}$ | $D_{[0.25,0.75]}$ | $\mathcal{E}_{\text{F}}$ |
| FQO-UNet | | | | | | | | |
| 10 % | 0.31 | 13.44 | [0.50, 1.00] | 6.6 | 0.32 | 14.75 | [0.50, 1.00] | 8.8 |
| 25 % | 0.19 | 9.63 | [0.60, 1.00] | 4.4 | 0.23 | 11.28 | [0.50, 1.00] | 6.6 |
| 50 % | 0.13 | 7.80 | [0.67, 1.00] | 3.5 | 0.17 | 9.13 | [0.60, 1.00] | 5.1 |
| 75 % | 0.11 | 6.67 | [0.67, 1.00] | 2.9 | 0.14 | 7.96 | [0.67, 1.00] | 4.6 |
| Full dataset | 0.09 | 5.99 | [0.74, 1.00] | 2.7 | 0.13 | 7.14 | [0.67, 1.00] | 4.6 |
| FQO-RN18 | | | | | | | | |
| 10 % | 0.44 | 18.33 | [0.33, 0.67] | 8.6 | 0.41 | 20.09 | [0.25, 1.00] | 11.1 |
| 25 % | 0.32 | 13.37 | [0.43, 0.67] | 6.1 | 0.32 | 14.68 | [0.50, 1.00] | 8.4 |
| 50 % | 0.23 | 10.76 | [0.60, 1.00] | 5.0 | 0.26 | 12.88 | [0.50, 1.00] | 7.3 |
| 75 % | 0.19 | 10.11 | [0.50, 1.00] | 4.0 | 0.22 | 10.93 | [0.50, 1.00] | 6.2 |
| Full Dataset | 0.17 | 8.58 | [0.60, 1.00] | 3.9 | 0.19 | 10.04 | [0.53, 1.00] | 5.6 |

Table 13: Test results for different training dataset sizes for both settings, V-5000 and G-5000.

### E.3 EVALUATION ON UNTRAINED FREQUENCIES

To test whether predictions generalize to frequencies, that were not part of the training data, we design the following experiment: We exclude the frequency range of 50 - 100 Hz during training and then evaluate the FQO-UNet on the whole frequency range. We report results on this in Table 14. We see that while the errors increase, the model is still able to make sensible predictions and even some peaks are predicted.

| Training condition | V-5000 | | | |
| --- | --- | --- | --- | --- |
| | $\mathcal{E}_{\text{MSE}}$ | $\mathcal{E}_{\text{EMD}}$ | $D_{[0.25, 0.75]}$ | $\mathcal{E}_{\text{F}}$ |
| Normal training | 0.04 | 0.87 | [1.00, 1.00] | 0.8 |
| 50 - 100 Hz excluded | 0.08 | 1.65 | [0.50, 1.00] | 2.0 |

Table 14: Evaluation only on 50 Hz - 100 Hz.

### E.4 MULTIPLE TRAININGS WITH RANDOM SPLITS

| Evaluation set | V-5000 | | | |
| --- | --- | --- | --- | --- |
| | $\mathcal{E}_{\text{MSE}}$ | $\mathcal{E}_{\text{EMD}}$ | $D_{[0.25, 0.75]}$ | $\mathcal{E}_{\text{F}}$ |
| Validation set | 0.095 (0.004) | 6.15 (0.23) | [0.69 (0.04), 1 (0)] | 2.79 (0.02) |
| Test set | 0.094 (0.002) | 6.00 (0.13) | [0.72 (0.02), 1 (0)] | 2.68 (0.05) |

Table 15: 6 models were trained on random splits in training and validation sets (4500 and 500 samples respectively). The results are denoted as mean (standard deviation).

