# OpenReview forum: "Vibroacoustic Frequency Response Prediction with Query-based Operator Networks"
_ICLR.cc/2024/Conference — Submitted to ICLR 2024_

### Official Review · Reviewer_4WC3 · 2023-10-23

**Soundness:** 4 excellent
**Presentation:** 4 excellent
**Contribution:** 4 excellent
**Rating:** 8
**Confidence:** 1

**Summary:**

This paper seems to introduce a novel benchmark dataset to explore vibrating plates frequency response, evaluate existing method on this new benchmark, and a propose a new architecture based on operator learning that outperforms existing methods in the literature.

Unfortunately, I do not feel capable to review this paper. I do not have a background on mechanics, physics, vibrating plates and acoustics; at the same time, I have never worked with operator learning, Fourier neural operators, and frequency domains in the context of artificial intelligence. Therefore, I lack the literature knowledge and experience from both the applied and methodological domain explored in this paper, making it impossible for my review to be useful for a proper evaluation at a conference like ICLR.

I have alerted the ACs to seek an opinion from a different reviewer.

----------------------
UPDATE: At the author's rebuttal, I was provided to an answer to what was a key weakness I've understood from the paper (seemingly only one test set). Instead, I was given results based on a cross validation procedure that I believe to be reasonable. Based on this, I change my rating from 6 to 8 (accept).

**Strengths:**

1. The code seems well organised overall, with a good folder/module structure.
2. The Related Work section seems specific and relevant for all the methods and applied area of the paper. It also seems to motivate well the paper.
3. The experiments (as, for example, seen in table 2) include one more "traditional" machine learning model (i.e, kNN). This is quite positive in the context of a area that sometimes forgets to compare new developments with simpler methods that might provide similar results. Although, from what I could understand in Appendix, this is based on an low-dimension embedding from a deep learning autoencoder model.
4. The paper provides an ablation analysis in Appendix.

**Weaknesses:**

1. The anonymous repository's documentation would benefit from some improvement. For example, it seems to be using Weights & Biases for logging but no explanation on how to use it is provided. Furthermore, the command in the README under "Train a model" section is not correct as the paths do not correspond to the folder structure. The setup script does not provide specific versions for the packages being installed (except for pytorch's cuda version) and, thus, this will impact the reproducibility of this work.
2. The experiment results are conducted in what it seems to be a single test set, and not through some sort of cross validation procedure to understand the variation and relevance of the results depending on the datasets' splits.
3. The datasets are synthetic, and do not seem to come from real-world measures. I understand from the paper that this is cost prohibitive, and that is why this new dataset is relevant, but I got the impression that this applied domain has implications in the real world, and therefore I'd guess and experiment in a real-world dataset, even if small, would be relevant.

**Questions:**

I do not have any questions.

---

> ### Author Response · Authors · 2023-11-16
>
> We appreciate your thoughtful review of our paper and are grateful for your positive remarks on the organization of our code, our Related Work section and on including a traditional machine learning model in our experiments.
>
> **Response to suggestions, concerns and questions:**
>
> 1. **Repository Documentation:**
>
> We acknowledge the issues you raised regarding our repository’s documentation. We will incorporate the suggested improvements by updating our readme with instructions, adding version numbers to the requirements.txt and adding an explanation for Weights and Biases setup.
>
> 2. **Single Test Set:**
>
> We follow the common paradigm in deep learning of splitting the data into training/validation and a single test set. To obtain an estimate of the variation we performed cross validation by six trainings with different randomly sampled training and validation splits for the FQO-UNet on the V-5000 setting. The statistics are presented in the following table.
>
> |Evaluation set|MSE|EMD|D[0.25, 0.75]|E_F|
> |---|---|---|---|---|
> |Validation set|0.095 (0.004)|6.15 (0.23)|[0.69 (0.04), 1 (0)]|2.79 (0.02)|
> |Test set|0.094 (0.002)|6.00 (0.13)|[0.72 (0.02), 1 (0)]|2.68 (0.05)|
>
> *Table: Six models were trained on random splits in training and validation sets (4500 and 500 samples respectively). The results are denoted as mean (standard deviation).*
>
> As we see, the variation across cross-validation runs is reasonably small. When evaluating on the fixed test set containing 1000 samples the variation decreases further.
>
> 3. **Synthetic Dataset and Real-World Application:**
>
> Real-world experiments are beyond the scope of this work due to prohibitive costs for manufacturing and measurements of field solutions (a measurement would require laser scanning vibrometry). However, previous research validated the use of the Mindlin model for the plate and errors of the finite element method are well understood in the engineering community [1].
>
>
> Please, let us know whether our rebuttal addresses your concerns and if there are further questions.
>
> ______
> [1]: Altenbach, Holm, et al. Analysis of Shells, Plates and Beams. Springer Nature, Switzerland, 2020.

---

> > ### Comment · Reviewer_4WC3 · 2023-11-17
> >
> > Thanks for the clarification. With regards to the single test set, now I see what you do makes sense and it was different from what I initially understood from the paper and it was a clear weakness to me. Based on your updated answer with the cross validation procedure, I'm increasing my rate to 8 (accept).

---

### Official Review · Reviewer_81Xi · 2023-10-30

**Soundness:** 3 good
**Presentation:** 3 good
**Contribution:** 3 good
**Rating:** 6
**Confidence:** 1

**Summary:**

Dear Area Chair,

I'm sorry that I'm not familiar with vibroacoustic wave propagation. Thus I'm not able to review this paper.

**Strengths:**

None.

**Weaknesses:**

None

**Questions:**

None

---

### Official Review · Reviewer_fgvw · 2023-11-01

**Soundness:** 4 excellent
**Presentation:** 4 excellent
**Contribution:** 4 excellent
**Rating:** 5
**Confidence:** 5

**Summary:**

This paper mainly proposes a PDE observer task that asks to predict the internal field information given the observation of plate geometry. The main contribution is the code and dataset, while it proposes a novel model of FQ-Operator. The proposed method outperforms the DeepONets and FNOs on the proposed dataset.

---

Post rebuttal comments: today, I held a long discussion with the AC to discuss this paper. The major concern is that this paper did not fit into the machine learning conferences very well. Senior ACs also suggest that authors should make efforts to help reviewers and the audience know the paper better. I wish to give the authors the following suggestions to improve their paper:

1. It is better to submit to a corresponding physics venue to find more suitable reviewers. I'm not an expert in wave propagation, and I cannot give a full assessment of this paper.

2. The title and abstract need to be revised to let machine learning people know what this paper is doing. For example, the title may be changed to "Query-based Operator Networks: Learning to Model Wave Propagation with Neural Operators".

3. In Table 1, kNN needs to be changed. kNN is a general methodology and it's better to be more specific.

4. In the introduction, 'geometry is represented with a neural network' is not rigorous in the eyes of a machine learning expert.

5. It's better to clarify the contribution of the proposed module against previous modules. More rationales need to be given.

6. It's better to work on a real physics dataset as well, or demonstrate the effectiveness of the proposed module on exisitng datasets.

7. It's better to introduce this work from a machine learning perspective, rather than focusing the task.

Based on above concerns, I need to decrease my score at this moment. However, I wish to encourage the authors, regardless of whether they wish to submit to ML conferences again or to physics journals.

**Strengths:**

Excellent paper!
1. I'm working on a similar task in this domain, and I appreciate and endorse the effort of this paper. My experience on these tasks convinced me this is a good application of machine learning in science. This paper will hold a unique position in this domain.
2. The construction of the proposed benchmark is well-motivated. I believe that the proposed frequency-domain problem is worth investigating. Also, I believe that this is also an important area for operator learning.
3. This paper is easy to follow, and many different lines of work in this domain are well-cited and discussed. This is another strong merit of this paper.
4. The proposed model is reasonable, and I believe it has some sort of novelty, although the major contribution of this work is in its formulations.
5. The limitations are well-discussed.

**Weaknesses:**

In general, I believe this paper's strengths outperform its weaknesses. And I believe this paper is certainly above the bar of this conference. My concerns are:
1. How easy the plate geometries can be acquired? The proposed beading patterns may not be easily gotten, and probably frequency or speed is probably easy to get. This discussion is required and is essential in this paper.

2. What's the distance between the simulations and the data collected in the real world? If the authors can conduct some experiments on real-world equipment, it would be very great.

3. The mesh geometries in this paper are not represented in vector form, and they are blurred. This should not happen and should be fixed in the rebuttal phase.

4. Why does Table 3 not have baselines' performances? It's better to test baseline methods on the transfer learning setup as well, and this is very critical.

5. It seems that the visualization of Figure 6 is not very good because (b) and (d) are not very similar. Also, I think it's better to add baseline comparisons or even videos (videos can be provided in the supplementary or the anonymous link).

6. The mechanical model should be put into the main text and linked to the dataset construction or methodology. This is very essential in the PDE domain.

7. What do grid-based models mean, and why is FNO not based on grids? I think FNO should have an option to work in regular grids. If grid-based methods are stronger, then FNO should also be put in the similar mode.

**Questions:**

Please refer to the above section for questions.

---

> ### Author Response · Authors · 2023-11-15
>
> We are deeply grateful for your evaluation of our paper and appreciate your recognition of its contribution to the field. Your endorsement, especially given your experience in this domain, is highly encouraging.
>
>
> **Response to suggestions, concerns and questions:**
>
> 1. **Plate Geometries Acquisition:**
> We acknowledge your concern regarding the practical aspects of producing plates with beading patterns. Adding beadings to plates to increase stiffness is an established method (see for example corrugations of tin cans) which can be conducted by stamping the plate. To actually produce the plates described in our work, stamps would have to be manufactured for the different beading patterns. The soft beading edges in our samples ensure manufacturability. We will include a more detailed discussion in the final manuscript.
>
>
> 2. **Real-World Data Simulations:**
> Our focus in this work is on establishing a data-driven benchmark dataset. Real-world experiments are beyond the scope of this work since special equipment for manufacturing and measurement would be needed. However, previous research provides good evidence for the validity of the Mindlin model and FEM in real world settings (see general answer). Generally, we are highly interested in real-world applicability and follow-up work should address this.
>
> 3. **Mesh Geometries Representation:**
> The mesh geometries are displayed (e.g. in Fig. 3) in their original discretized (an $81 \times 121$ grid). We did not apply interpolation or other image improvements to have a faithful presentation. This will be explained in the figure caption.
>
>
> 4. **Baseline Performances in Table 3:**
> We agree that baselines are important and follow your suggestion of adding new baselines to the revised manuscript for a more comprehensive comparison of transfer learning performance.
>
>
> 5. **Visualization in Figure 6:**
> We appreciate your feedback on Figure 6 and suggestions for further visualizations. Regarding Figure 6, we want to make the point that despite visible errors in the field solution prediction, the frequency response (=average) is well estimated.
> We agree that providing more visualizations of the model results is a good idea and will add them. For now, we provide several gifs showcasing the predicted and ground truth field solutions of the FQO-UNet.
> https://drive.google.com/drive/folders/1EUeeAtPm0ZFdC6oIBsxJQbsOhhfXVUk4?usp=sharing
>
> 6. **Mechanical Model in Main Text:**
> We agree on the importance of linking the mechanical model to our methodology and dataset construction. This will be changed in the revised manuscript.
>
> 7. **Grid-Based Models and FNO:**
> By grid-based models we refer to models, that can only make predictions for predefined, equidistant frequencies (i.e. a 1-dimensional grid). As you correctly point out, this holds for our implementation of the FNO method as well. The Grid-UNet and Grid-RN18 are ablation variations of the FQO-UNet and FQO-RN18, where we removed the frequency query mechanism.
> We understand that the distinction between baseline methods and the ablation variations of our proposed method needs to be clarified. We will do so in our updated manuscript.
>
> Please let us know, if there is still something left unclear!

---

> > ### Comment · Reviewer_fgvw · 2023-11-15
> > **Almost empty file in solution_1.gif**
> >
> > Thanks for your response. I noticed that most frames in solution_1.gif look empty (all dark) in prediction and in the actual field solution. Is there anything wrong?

---

> > > ### Author Response · Authors · 2023-11-15
> > > **This is due to scaling.**
> > >
> > > Thanks for your question.
> > >
> > > The plots show the field solution and prediction in m/s and are scaled with respect to the maximum and minimum values in the solution across all frequencies. Because of this, for some frequencies the velocity is too small to be visible. We added a folder called gifs_no_scaling with new gifs. Here, each frame is scaled individually and more details are visible.

---

> ### Comment · Reviewer_fgvw · 2023-11-16
> **Thanks, it's now visible.**
>
> It's, in general, a good pioneering work. I'll discuss this with other reviewers, and if no further concerns are raised, I'll keep my score unchanged.

---

### Official Review · Reviewer_K93H · 2023-11-01

**Soundness:** 3 good
**Presentation:** 3 good
**Contribution:** 3 good
**Rating:** 5
**Confidence:** 2

**Summary:**

This paper introduces a dataset of 12,000 plate geometries and their associated frequency responses for a frequency prediction problem. A metric is introduced to evaluation the quality of the response. An operator model is also introduced for solving the frequency prediction problem. Numerical results show the competitiveness of the model.

**Strengths:**

- This paper focuses on the frequency response prediction problem, which currently has no structured benchmark
- The data is simulated based on a plate theory differential equation, and solved using FEM which has theoretical guarantees on its convergence
- A throughout literature review on related work is given
- Overall, it is a clearly written paper and easy to follow
- The limitations of the paper are discussed

**Weaknesses:**

- It seems that in the numerical experiments, there is no noise added to the data. Predicting data without noise is not a difficult task from a data-fitting perspective, especially with large models like deep networks.
- The details of the finite element method and linear solver are missing, which is crucial to the generation of the data
- It seems that in the calculation of the aggregation frequency response, the paper only describes how it is done, but no justification is given. Are there any references which can justify the calculation steps?
- In the results, it seems that the the sizes (e.g., number of parameters) of the model are not reported. This is important because the proposed model is outperforming because it uses larger networks. Also, it does not mention how the parameter tuning is performed.

**Questions:**

- In the dataset generation, the displacement is set to 0, which means that a term in the plate theory differential equation is set to 0. Does it mean that the plate is static? Is it a common setting in existing work?
- Also, it assumes that the vibration comes from a single point. Is this setting also used in related work? Are there any references to it?
- The data is generated by solving the differential equation use FEM and a linear solve to (1). Thus, the accuracy of the solution is crucial here. What is the error to equation (1) after applying the linear solver? How well does the generated data satisfy the differential equation?
- How does the proposed method perform on frequencies that it is not trained on? Since one strength of the proposed operator model is that it can be evaluated at any frequency values, it is important to show that it can empirically generalize to other frequencies well.

---

> ### Author Response · Authors · 2023-11-15
>
> We thank you for your recognition of our paper's contributions. Your positive feedback on our paper’s problem, finite element methodology, literature review, clarity, and discussion of limitations is greatly appreciated.
>
> **Response to suggestions, concerns and questions:**
>
> 1. **Absence of Noise in Data:**
> Our work concerns learning models, that predict the frequency response of a vibrating plate. The primary goal is to learn a faithful surrogate model. All experiments evaluate scores on the test set, which consists of unseen plate geometries. Hence, they measure generalization and not just how well the data is fitted. Nonetheless, we agree that robustness to noise is important and particulary relevant for real world transfer (for example when manufacturing is imperfect) and should be addressed by follow-up work.
>
> 2. **Details for Finite Element Method and Solver:**
> We agree that the details of the finite element method and linear solver are crucial. We employ a specialized FEM implementation for acoustics [1]. The setup involves shell elements with 9 nodes, 6 degree of freedom and quadratic ansatz functions for discretization. The direct solver MUMPS [2] is employed. More comprehensive descriptions will be included in the final manuscript.
>
> 3. **Aggregated Frequency Response Justification:**
> Calculating the aggregated frequency response by taking the mean squared velocity is a common choice in vibroacoustic simulation. The mean squared velocity is an important measure for the radiated sound power. Therefore, acoustic engineers use this quantity to evaluate structural design measures [3]. We will clarify this in the paper.
>
> 4. **Model Size and Hyperparameter Tuning:**
> We agree on the relevance of model and training details. Model specifications are reported in Appendix C and ablation studies that document our hyperparameter tuning are listed in Appendix D. We will add the number of parameters for the different models to Appendix C. In our experiments, naively increasing the model size did not lead to better performance.
>
> 5. **Question: Dataset Generation Settings:**
> Thank you for highlighting this aspect. In our dataset, the zero displacement condition is applied exclusively at the plate's edges. This means the central part of the plate can move (vibrate) freely, while the edges are fixed. This modeling choice is common in structural analysis, especially in engineering applications where plates are often components of more complex assemblies. To prevent further misunderstandings, we clarify this in our manuscript.
>
> 6. **Single-Point Vibration Assumption:**
> A point force excitation is one scenario among many (e.g. spatially distributed load or plane wave excitation). There are good reasons to choose a point force excitation: In real world applications, a point force can represent the force application point commonly defined by the bearing of a structure. Additionally, point actuators are used to excite the structure for experimental validation. Therefore, this setting is often employed in vibroacoustics, see e.g. [4] and [5]. We also choose an asymmetric excitation point to excite the important Eigenmodes. We will clarify this in the paper.
>
> 7. **Accuracy of FEM and Solver:**
> Generally, the finite element approach in combination with linear direct solvers converges to the exact solution [6]. The accuracy of our solutions in relation to equation (1) is assured by a suitable discretization. In particular, we have chosen the mesh size such that the smallest occurring bending wavelength is discretized with at least 10 nodes. The smallest bending wavelength occurs at the highest frequency and can be analytically calculated for thin plate structures. This approach is common practice for choosing the mesh size of vibroacoustic models [7].
>
> 8. **Performance on Untrained Frequencies:**
> We recognize the importance of our model's generalization capabilities, which we've explored for different amounts of beadings as detailed in our transfer learning section. Extending this to unseen frequency ranges is a great idea! We conducted this experiment and present our results in the general answer.
>
> We would be interested to hear, whether our rebuttal addressed your concerns.

---

> > ### Author Response · Authors · 2023-11-15
> > **References**
> >
> > [1]: Sreekumar, H and Langer, S. C.: "Elementary parallel solver core module for high performance vibroacoustic simulations", 2023. DOI:10.24355/dbbs.084-202301301305-0
> >
> > [2]: Amestoy, Patrick R., et al.: "MUMPS: a general purpose distributed memory sparse solver", International Workshop on Applied Parallel Computing. Springer, 2000.
> >
> > [3]: Sung, C-C., and Jan, J. T.: "The response of and sound power radiated by a clamped rectangular plate", Journal of sound and vibration 207.3: 301-317. 1997.
> >
> > [4]: Yairi, M. et al.: "Sound radiation from a double-leaf elastic plate with a point force excitation: Effect of an interior panel on the structure-borne sound radiation", Applied Acoustics 63.7: 737-757, 2002.
> >
> > [5]: Shorter, P. J., and Langley, R. L.: "Vibro-acoustic analysis of complex systems", Journal of Sound and Vibration 288.3: 669-699, 2005.
> >
> > [6]: Atalla, N., and Sgard, F.: "Finite element and boundary methods in structural acoustics and vibration", CRC Press, 2015.
> >
> > [7]: Marburg, S. "Discretization requirements: How many elements per wavelength are necessary?", Computational Acoustics of Noise Propagation in Fluids-Finite and Boundary Element Methods: 309-332. Springer, 2008.

---

> > > ### Author Response · Authors · 2023-11-21
> > > **Respectful reminder**
> > >
> > > We would be grateful if you took the time to take a look at our rebuttal and revised manuscript. We are interested in whether our rebuttal addressed your concerns and your valuable feedback.

---

### Official Review · Reviewer_SgdV · 2023-11-02

**Soundness:** 3 good
**Presentation:** 3 good
**Contribution:** 2 fair
**Rating:** 6
**Confidence:** 3

**Summary:**

This paper has two contributions: a dataset of vibroacoustic frequency responses from metal plates and learning models addressing the problem of predicting such responses. Numerical computation of such responses is expensive, so a learning model may be able to approximate the computation and directly predict responses at different frequencies. The dataset contains 12k plate geometries and their responses computed from the numerical method. The work has also proposed a list of evaluation metrics for a prediction model. It provides two models and tests them on the datasets.

**Strengths:**

The problem is interesting. It represents another application that uses neural networks to approximate numerical computations.

The model choices are also reasonable. It uses convolutional neural networks or ViT to extract a compact representation of the plate and then make predictions from there. The experiment results are sufficient.

**Weaknesses:**

Considering that this work is applicational and has little contribution to methodology, I do worry that this work does not have enough audience in the ICLR community.

**Questions:**

How do you distinguish different methods as baselines and proposed methods? I feel that they are all applications of existing models with minor changes.

---

> ### Author Response · Authors · 2023-11-15
>
> Thank you for your insightful review of our paper. We are grateful for your recognition of the interesting nature of the problem we address, the adequacy of our model choices, and the sufficiency of our experimental results.
>
> **Response to suggestions, concerns and questions:**
>
>
> 1. **Concerns on Audience and Contribution:**
> We would like to emphasize the position of our research in the established fields of *Machine learning for physics and science*. While machine learning applications to engineering are a fairly new topic in the machine learning community, there is great potential for impactful research. Applications of machine learning to other domains were highly successful, e.g. AlphaFold in biology [1]. Our work seeks to bridge the gap between the communities by challenging data-driven approaches with a mechanical engineering problem. While most existing research models the temporal evolution of dynamic systems, our work's novelty is to directly predict solutions in the frequency domain. To address this task, we contribute a benchmark dataset, scores of relevant baselines, and the new FQO-UNet model.
>
>
> 2. **Question on Baselines vs. Proposed Method:**
> As baselines, we selected methods that were proposed in the literature for related physics-based problems. In contrast, our proposed FQO-UNet is specifically designed to address the unique challenges of vibroacoustic frequency response prediction.
> For instance, the prediction of resonance peaks is addressed by the frequency-query approach. However, we agree that the distinction between baseline methods and the ablation variations of our proposed method needs to be emphasized and we will do so in our updated manuscript.
>
> We would be grateful if you report whether our answer addressed your concerns or if additional questions have come up.
>
> ---------------------
> [1] Jumper et al.: "Highly accurate protein structure prediction with AlphaFold", Nature (2021)

---

> > ### Author Response · Authors · 2023-11-21
> > **Respectful reminder**
> >
> > We would be grateful if you took the time to take a look at our rebuttal and revised manuscript. We are interested in whether our rebuttal addressed your concerns and your valuable feedback.

---

> > > ### Comment · Reviewer_SgdV · 2023-11-23
> > > **Thank you for your clarification**
> > >
> > > Thank you for your clarification.
> > >
> > > As machine learning conferences grow larger and larger, I don't feel that all work related to machine learning should be put into these conferences. The target audience of this work is unlikely to be researchers in machine learning, so I am not sure ICLR should be the venue for the work to generate impact.

---

### Author Response · Authors · 2023-11-15
**General Response**

We thank all reviewers for their valuable comments, suggestions and the generally positive judgement, especially for acknowledging: the problem to be relevant (SgdV, K93H, fgvw), reasonable models choices (SgdV, fgvw), good discussion on related work (K93H, fgvw, 4WC3), novelty (K93H,  fgvw), benchmark experiment design (SgdV, fgvw, 4WC3) and good readability (K93H, fgvw).

We will upload the revised version of our manuscript soon. We present the following additions suggested by the reviewers:

1. Video visualization of dataset and predictions
2. Explanations regarding real world applicability
3. Evaluation on unseen frequencies


**1. Video Visualization of Dataset and Predictions**
Reviewer fgvw kindly suggested to visualize the dataset as videos. For now, we provide several gifs showcasing the predicted and ground truth field solutions of the FQO-UNet.  Note, the plots in the folder "gifs" are scaled with respect to the maximum and minimum values in the solution across all frequencies. Because of this, they have a consistent scale for comparison, but some details are not visible. The frames for the plots in the folder "gifs_no_scaling" are scaled individually, so more details are visible.

https://drive.google.com/drive/folders/1EUeeAtPm0ZFdC6oIBsxJQbsOhhfXVUk4?usp=sharing

**2. Background: Real World Applicability / Finite Element Model Validation**

* **Validation of the physical model:** The Mindlin plate theory is widely applied in engineering [1] to model the dynamical response of moderately thin structures in a low to mid frequency range. Early works [2] have validated the Mindlin plate model against experimental data by comparing the predicted and measured resonance frequencies and found good agreement.

* **Verification of the FEM implementation:** We have to ensure that the bending waves are well resolved. This requires a minimum of 10 nodes per wavelength. The minimum bending wavelength occurring at the maximum frequency can be calculated. In our model, we have chosen an even finer discretization to accurately resolve the geometry variations of the beading patterns. We have added the calculation of the minimum bending wavelength to the appendix.

We acknowledge that these points should be motivated well and will update the paper accordingly.

**3. Evaluation on Untrained Frequencies**
Our FQO-UNet method can be queried for frequencies that have not been seen during training. To test whether the predictions made without any finetuning on new frequencies show some kind of generalization we design the following experiment. We exclude the frequency range of 50 - 100 Hz from the training set and then evaluate the FQO-UNet on the whole frequency range. We report preliminary results on this in the table below. We see that while the errors increase, the model is still able to make sensible predictions and even some peaks are predicted.

| |MSE|EMD|D[0.25, 0.75]|E_F|
|-|-----|---|---|---|
|Normal training|0.04|0.87|[1.00, 1.00]|0.8|
|50-100 Hz excluded|0.08|1.65|[0.50, 1.00]|2.0|

*Evaluation only on 50 Hz - 100 Hz.*


**Discussion:**
We would like to have further exchange with you. Feel free to ask more questions!

----------------------
[1]: Altenbach, Holm, et al. Analysis of Shells, Plates and Beams. Springer Nature, Switzerland, 2020.

[2]: Russell, David L., and Luther W. White. "Formulation and validation of dynamical models for narrow plate motion." Applied mathematics and computation 58.2-3 (1993): 103-141.

---

> ### Author Response · Authors · 2023-11-18
> **Revision uploaded**
>
> We thank all reviewers for their constructive feedback. Following your suggestions, we have uploaded an updated version of our paper. The additions have been marked red for clarity. We have also added new videos comparing the field solution predictions of different methods as supplementary material and to the google drive folder.

---

### Meta-Review · Area_Chair_4JCr · 2023-12-11

**Metareview:**

The paper introduce a data-driven approach to simulate vibroacoustic wave propagation in airplanes, cars or some mechanical structure.
Instead of using expensive numerical solution based on finite element solvers, the authors introduce a synthetic dataset that includes various geometries. From a machine learning point of view, it seems to follow work on learning ODE/ PDE solvers but applied to frequency domain data. Here the mechanical models comes from a classical equation introduced in the 50s and whose (considered ground-truth) solution are obtained with finite-element discretization solvers.

The details to understand how this is used is quite unclear. The paper seems to be written for people already very familiar with the problem and the ML part is poorly described.


> The paper is hard to assess

The large majority of the reviewers expressed discomfort assessing this paper. The background to understand the core contribution are lacking and the paper significance is difficult to assess.

**Justification For Why Not Higher Score:**

Without confident reviews, I can hardly argue for acceptance of this paper. I believe that if it didn't find interest among the reviewers, the current state of the paper might not find public in the conference. It might be unfortunate in case where it contains nice contributions, but the paper is hard to read and need some deep revisions before acceptance.

**Justification For Why Not Lower Score:**

N/A

---

### Decision · Program_Chairs · 2024-01-16

Reject